# Greater Circulating Copper Concentrations and Copper/Zinc Ratios are Associated with Lower Psychological Distress, But Not Cognitive Performance, in a Sample of Australian Older Adults

**DOI:** 10.3390/nu11102503

**Published:** 2019-10-17

**Authors:** Michelle Mravunac, Ewa A. Szymlek-Gay, Robin M. Daly, Blaine R. Roberts, Melissa Formica, Jenny Gianoudis, Stella L. O’Connell, Caryl A. Nowson, Barbara R. Cardoso

**Affiliations:** 1Institute for Physical Activity and Nutrition (IPAN), School of Exercise and Nutrition Sciences, Deakin University, Geelong, VIC 3220, Australia; michelle.mravunac@eweb.endeavour.edu.au (M.M.); ewa.szymlekgay@deakin.edu.au (E.A.S.-G.); robin.daly@deakin.edu.au (R.M.D.); melissa.prosser@deakin.edu.au (M.F.); j.gianoudis@deakin.edu.au (J.G.); stella.oconnell@deakin.edu.au (S.L.O.); caryl.nowson@deakin.edu.au (C.A.N.); 2Nutrition Society of Australia, PO Box 576, Crows Nest, NSW 1585, Australia; 3Melbourne Dementia Research Centre, The Florey Institute of Neuroscience and Mental Health, Parkville, VIC 3052, Australia; blaine.roberts@florey.edu.au; 4Department of Nutrition, Dietetics and Food, Monash University, Notting Hill, VIC 3128, Australia

**Keywords:** zinc, copper, copper/zinc, cognition, dementia, depression, anxiety, neurotrophic factors

## Abstract

Dyshomeostasis of copper and zinc is linked to neurodegeneration. This study investigated the relationship between circulating copper and zinc and copper/zinc ratios and cognitive function, symptoms of depression and anxiety, and neurotrophic factors in older Australian adults. In this cross-sectional study (*n* = 139), plasma copper, serum zinc, and neurotrophic factors (brain-derived neurotrophic factor (BDNF), vascular endothelial growth factor, and insulin-like growth factor-1) were assessed. Cognition was assessed using the Cogstate battery and the Behavior Rating Inventory (BRI) of Executive Function (Adult version). Symptoms of anxiety and depression were assessed with the Hospital Anxiety and Depression Scale. Copper (β = −0.024; 95% CI = −0.044, −0.004; *p* = 0.019) and copper/zinc ratio (β = −1.99; 95% CI = −3.41, −0.57; *p* = 0.006) were associated with lower depressive symptoms, but not cognition. Plasma copper had a modest positive association with BDNF (β = −0.004; 95% CI = 0.000, 0.007; *p* = 0.021). Zinc was not associated with any of the outcomes. In conclusion, greater circulating copper concentrations and higher copper/zinc ratios were associated with lower depressive symptoms (but not cognition), with copper also positively associated with BDNF concentration, in a sample of community-dwelling older adults.

## 1. Introduction

Ageing is a naturally occurring physiological process which is characterized by both physical and cognitive decline. Changes are seen in brain neurochemistry as well as in neuroanatomical structures, which negatively affect synaptic plasticity and contribute to atrophy [1,2]. Ageing is also associated with increased vulnerability of the neurons to oxidative stress, emphasizing the potential importance of changes in the concentration of neurotrophic factors such as a decrease in the concentrations of brain-derived neurotrophic factor (BDNF), which supports the growth and maintenance of neurons [3]. Decreased concentrations of such neurotrophic factors render brain cells more prone to damage caused by inflammatory processes and oxidative stress [4,5], which are key events in age-associated cognitive decline and neurodegenerative disease pathogenesis [6] experienced by many older adults. In fact, it is estimated that dementia, a far-end result of cognitive decline, affects approximately 50 million people worldwide [7]. This number is expected to increase to 75 million by 2030 and 132 million by 2050, posing a considerable burden on the health care system [7]. Dementia is intrinsically linked with depression, as both are characterized by neuroinflammation and oxidative stress [8]. Depression is a complex disease involving a chronic low-grade inflammatory response, activation of cell-mediated immunity, and progressive neuropathology [9], which can lead to neurodegeneration and later on contribute toward the development of dementia [10]. Besides being one of many risk factors for dementia [11], depression is estimated to be the leading cause of disability by 2030 [12].

Metal ions such as copper and zinc play an essential role in brain function and neuronal homeostasis [13], and long-term imbalance of these metals has been linked to neurodegeneration and neurological disorder [14]. It is thought that this may be due to the provocation of an unchecked immune response within the brain via the increased production of reactive oxygen species [15], which can lead to brain damage via the upregulation of inflammatory mediators [16]. Copper is an essential transition metal that acts as a catalyst for several biological processes such as the synthesis of neurotransmitters. However, disturbance of its transport, causing accumulation in the blood and depletion in the brain, has been linked with neurodegenerative diseases such as Alzheimer’s, the most common form of dementia [17]. Copper works synergistically with zinc in several metalloenzymes, such as the antioxidant enzyme superoxide dismutase. Zinc is an essential component of numerous biochemical and molecular processes related to cell survival and, unlike copper, has very low toxicity in humans [18]. However, research suggests that zinc deficiency may increase the risk of dementia [19,20], and thus supplementation with zinc may improve cognition and regulation of emotional behavior (such as anger and depression) in cognitively intact individuals [21].

Zinc and copper are also likely to have a synergistic role in Alzheimer’s disease, with copper promoting oxidative stress and displacing zinc from body tissues [18,22]. Therefore, not only is the individual nutritional status of copper and zinc of concern, but the balance between these two metals appears to be important. Indeed, studies suggest that calculating the copper to zinc ratio (Cu/Zn ratio) provides valuable information in regard to oxidative stress and inflammatory responses [23]. Ageing is associated with decreased zinc status and hence increased Cu/Zn ratio, which is also observed in individuals with dementia [24]. Therefore, this study aimed to investigate the relationship between plasma copper and serum zinc concentrations, as well as the Cu/Zn ratio, neurotrophic factors, cognitive function, and psychological distress in a sample of Australian community-dwelling older adults.

## 2. Materials and Methods

### 2.1. Study Design

This cross-sectional study is a secondary analysis of baseline data from the Seniors’ Thinking, Exercise and Protein Study (STEPS), a randomized controlled trial (RCT) that aimed to assess the effects of an exercise program combined with a protein-enriched diet on skeletal muscle mass, strength, size, and cognitive function in community-dwelling older Australian adults [25]. This study was conducted within the Institute for Physical Activity and Nutrition at Deakin University, and according to the guidelines laid down in the Declaration of Helsinki. All procedures involving human subjects were approved by the Deakin University Human Research Ethics Committee, Australia (project number: 2013-166). Written informed consent was obtained from all participants prior to their inclusion in the study. The trial was registered at the Australian New Zealand Clinical Trials Registry (registered at www.actr.org.au as ACTRN12613001153707). 

### 2.2. Participants 

A total of 154 community-dwelling men and women aged ≥60 years were enrolled in the study. Participants were recruited from local community groups in Melbourne and surrounding areas in Victoria, Australia, by local media campaigns, presentations at local community groups, and word of mouth. Interested participants were screened over the telephone and excluded based on the following criteria [25]: Current or prior participation (past 3 months) in ≥150 min per week of moderate-intensity physical activity or resistance exercise >1 day per week; illness which was acute or terminal, including current cancer or surgery and cessation of radiotherapy/chemotherapy less than 12 months before the study; functional impairment which would limit participation in the exercise program; use of insulin to treat diabetes; body mass index (BMI) >40; presence of chronic liver disease, coeliac disease, ulcerative colitis or Crohn’s disease; use of oral corticosteroids in the past 6 months; or inability to commit to the study and its requirements. Eligible participants were screened for depression with the Geriatric Depression Scale (GDS) [26] and for dementia with the Short Portable Mental Status Questionnaire (SPMSQ) [27]. Participants with a score of >6 on the GDS or a score of >2 on the SPMSQ were excluded from the study. For this cross-sectional analysis, 15 participants were excluded as they had missing data for plasma copper, serum zinc or the apolipoprotein E (APOE) genotype. Thus, a total of 139 participants were included in this study.

### 2.3. Demographics and Clinical Assessment

Participants provided socio-demographic, lifestyle, and history of disease(s) information via a self-administered questionnaire. Education was categorized as completion of primary/high school, technical certificate or university. History of cardiovascular disease (CVD) was defined as self-reported history of heart attack, stroke, or heart disease. Habitual physical activity time was estimated for the preceding four weeks based on self-reported information using the Community Healthy Activities Model Program for Seniors (CHAMPS) physical activity questionnaire [28]. Height was measured with a wall-mounted stadiometer to the nearest 0.1 cm, and body weight was measured using calibrated electronic digital scales to the nearest 0.1 kg. Body mass index (BMI) was derived from weight (kg) divided by squared height (m^2^).

### 2.4. Blood Samples

Participants attended a commercial pathology clinic with multiple collection centers in Melbourne, where a fasted, morning venous blood sample was collected. Blood was collected into an EDTA vacutainer (to obtain whole blood and plasma after processing) and a trace element-free vacutainer (to obtain serum after processing). Plasma and serum were separated by centrifugation at 3000× *g* for 15 min at 4 °C using trace element-free polyethylene disposable pipettes. Samples were stored in trace element-free polyvials at −80 °C until analysis.

### 2.5. Copper and Zinc Concentrations

Plasma copper concentration was measured by inductively coupled plasma mass spectrometry (ICP-MS) (7700x system, Agilent Technologies, Melbourne, Australia). Accuracy of the assay was assessed using reconstituted lyophilised Seronorm™ Trace Elements in Serum (Sero AS, Billingstad, Norway) as the standard reference material, which was prepared using the same protocol as for the plasma samples. The measured analytical recovery of copper in the Seronorm™ standard was within the acceptable range, as per manufacturer’s guidelines (measured seronorm = 153 (SD 28) µg/dL; certified range = 170–200 µg/dL).

Serum zinc concentration was measured via flame atomic absorption spectrophotometry (AAS) (Varian SpectrAA-800, Varian Inc., Palo Alto, CA, USA) by direct aspiration. The precision of the serum zinc analysis was checked using a pooled human serum sample, and the accuracy of the method was checked using a serum UTAK certified reference control (66816 Normal Range, Lot Number A4913). The measured analytical recovery of zinc in the UTAK control was within the acceptable range (measured UTAK = 60.8 (SD 2.8) µg/dL; certified range = 51.8–70.2 µg/dL). Because serum zinc has been shown to be influenced by inflammatory status [29], zinc concentrations were adjusted for interleukin (IL)-6, an inflammatory marker that reflects both acute and chronic inflammatory states, by using a regression correction approach [29]. As the first step of this approach, a reference IL-6 value was determined by the maximum value of the lowest decile of its log-transformed values. Then, this reference value was subtracted from the observed IL-6 values to determine the adjusted IL-6. A linear regression model was developed with zinc as the dependent variable and adjusted IL-6 as the independent variable. The coefficient (β) obtained from the regression model was then used to adjust for the effect of inflammation on zinc using the following formula: Adjusted zinc = raw zinc − β (adjusted IL-6) [29]. Adjusted zinc values were used to calculate the Cu/Zn ratio. 

IL-6 was measured with Milliplex T Cell high-sensitivity human cytokine panel (Millipore Billerica, MA, USA) as per manufacturer’s recommendations, with an intra-assay coefficient of variability (%CV) of 5.9–11.7% and an inter-assay %CV of 7.3–15.7%.

### 2.6. Neurotrophic Factors Concentration in Serum

Neurotrophic markers BDNF and vascular endothelial growth factor (VEGF) were determined by using the commercial DuoSet ELISA kit (R & D Systems, Minneapolis, MN, USA), with an intra-assay %CV of 3.9–5.9% and inter-assay %CV of 4.4–14.7%. Insulin-like growth factor 1 (IGF-1) was measured by using the Immulite 2000 IGF-1 chemiluminescent immunometric assay (Siemens Healthcare Diagnostics, Los Angeles, CA, USA), with an intra-assay %CV of 3.1 and inter-assay %CV of 6.2.

### 2.7. APOE Genotyping

APOE genotype was assessed in whole blood using a polymerase chain reaction (PCR)-based assay, designed with the MassARRAY Assay Design 4.0 software (Agena Bioscience, San Diego, CA, USA). The initial PCR step involved 45 cycles (annealing temperature of 56 °C), and the PCR products were treated with shrimp alkaline phosphatase (15 min at 37 °C) and denatured at 85 °C for 5 min. The iPLEX extension step had 40 cycles of lots of five cycles (between 52 °C and 85 °C). The resulting iPLEX extension products were desalted using SpectroCLEAN resin (SEQUENOM, San Diego, CA, USA) and then spotted on SpectroCHIPs GenII (SEQUENOM, San Diego, CA, USA) for analysis with the MassARRAY Analyser Compact MALDI-TOF MS (Agena Bioscience, San Diego, CA, USA). In this study, APOE carrier status was defined by the presence (1 or 2 copies: ε4 carrier) or absence (0 copies: non-ε4 carrier) of the APOE ε4 allele.

### 2.8. Cognitive Function and Depression Symptoms

Cognitive performance was assessed using the Cogstate computerised cognitive tests [30], which provide a valid and sensitive measurement of a range of different cognitive functions, such as attention, memory, processing speed and executive function, and have been validated in older adults [31,32]. All cognitive assessments were undertaken on a laptop with a mouse and headphones. As reported previously [25], written and verbal instructions were provided, and participants were allowed a practice trial prior to completing five cognitive tasks. Briefly, the Cogstate tests used in this study are based on game-like playing card stimuli and tasks that measure simple reaction time and psychomotor function (Detection, DET), choice reaction time and visual attention (Identification, IDN), visual recognition memory and attention (One card learning task, OCL) and working memory and attention (One-back task, ONB). Additionally, participants completed a maze-like task (Groton Maze Learning, GML) that provides a measure of executive function, memory and spatial problem-solving. DET, IDN and ONB were scored using reaction time (in milliseconds); accuracy in OCL was scored using the number of correct responses; and the GML task was scored using the total number of errors on five consecutive trials at a single session [31,32]. Scores for DET and IDN were normalized using a log10 transformation; OCL and ONB scores were normalized using an arcsine square-root transformation. Raw scores were transformed into a *z*-score using the mean and standard deviation of the total sample in the study. Three composite scores were computed from these five tests [30]: i) Working memory/learning: Corresponding to the average *z*-scores for OCL and ONB; ii) attention/psychomotor function: Corresponding to the average of the *z*-scores for DET and IDN; and iii) global cognitive function: Represented by the average of the *z*-scores for all five tasks.

A self-report of executive function was also completed using the Behavior Rating of Inventory Executive Function—Adult version (BRIEF-A) [33]. This is a self-report comprised of 75 questions that measure various aspects of executive functioning, and are made up of nine non-overlapping theoretically and empirically derived clinical scales. These scales include: inhibit, self-monitor, plan/organise, shift, initiate, task monitor, emotional control, working memory, and organization of materials [33]. Each question had the option of a response of never, sometimes or often. This was then recorded for how often each listed behavior had been a problem in the last month. Each response corresponds to a numerical score of 1 to 4, and these were summed to provide scores for each subdomain. The inhibit, shift, and emotional control subdomains were summed to provide a behavioral regulation index (BRI) score, and the other subdomains were summed to provide a metacognition index score (MI). Lastly, the BRI and MI were summed to provide an overall global executive composite (GEC) summary score [33]. For the results, T-scores were derived for each scale, with higher scores representing a greater degree of executive dysfunction and levels of impairment.

Anxiety and depression symptoms were assessed with the Hospital Anxiety and Depression Scale (HADS) [34]. The HADS questionnaire is a self-reported tool to measure psychological distress in non-psychiatric patients, and has been widely used in community populations [35,36]. The HADS rating scale consists of 14 items on a four-point Likert scale (each item is scored 0–3 where a higher score represents more severe depression or anxiety), which includes seven questions assessing the emotional and cognitive aspects of depression, and seven questions evaluating the emotional and cognitive aspects of anxiety. The results are reported as the subscales Depression and Anxiety comprised of seven items each (maximum 21). Scores of 11 or more on either subscale are considered to indicate the probable presence of psychological morbidity, while scores of 8–10 are indicative of possible mild morbidity, and 0–7 represent normal psychological state [34].

### 2.9. Statistical Analysis

All statistical analyses were performed with STATA/SE 15.0 for Windows (StataCorp LLC, College Station, TX, USA). Data were presented as mean ± standard deviation (SD) if normally distributed, and as median (25th, 75th percentile) if not normally distributed according to the Shapiro Wilk test. Categorical variables were presented as *n* (%). Multiple linear regression models were used to examine the relationship between cognitive outcomes and the metals (plasma copper, serum zinc, and Cu/Zn ratio) adjusted for the following covariates: Sex, age, BMI, APOEε4 status, education level, history of cardiovascular disease, and habitual moderate–vigorous physical activity. Linear regression models were also used to investigate the association between metals and neurotrophic factors. These models were adjusted for the same covariates, except for education. We tested whether there were any sex interactions for the associations between cognitive outcomes and neurotrophic factors (dependent variables) with the metal markers in multivariable models adjusted for the same covariates as described above, and no significant interactions were observed. In order to meet statistical assumptions of normality and homogeneity of variance, neurotrophic factors (BDNF, IGF-1, and VEGF) were log-transformed for inclusion in regression analyses. There was no correction for multiple comparisons to minimize type II error, and a *p*-value < 0.05 was considered significant. Considering this is a secondary analysis, power calculations were not performed prior to the study. However, an F test was calculated for the coefficient of determination (R^2^) considering the lowest significant R^2^ observed in our findings (0.07), and the estimated power is of 0.61.

## 3. Results

Table 1 shows the demographic characteristics of participants included in this study. The study population was comprised mostly of women (63%), with an average age of 70.7 years (range 65–84 years), and 21% were classified as APOEε4 carriers. In this study population, 15.8% of men (*n* = 22) were zinc deficient (serum concentration < 74 µg/dL) [37], and 22.3% of women (*n* = 31) were zinc deficient (serum concentration < 70 µg/dL) [37].

Multiple regression analysis showed no significant association between copper and Cu/Zn ratio and anxiety symptoms or cognitive performance (Figure 1, Table 2). However, higher plasma copper concentrations and higher Cu/Zn ratio were associated with lower levels of psychological distress as assessed by the HADS Depression scale (Figure 2, Table 2). Serum zinc concentration was not associated with any of the cognitive outcomes or HADS subscales (Figure 1 and Figure 2, Table 2). 

Multiple linear regression analysis also revealed a modest positive association between plasma copper concentrations and BDNF, while serum zinc and Cu/Zn ratio were not associated with any of the neurotrophic factors (Table 3).

## 4. Discussion

The main finding from this study was that in a sample of community-dwelling older adults, greater plasma copper, as well as Cu/Zn ratios, were associated with lower depressive symptoms. Furthermore, a modest association between plasma copper concentration and the neurotrophic factor BDNF was observed in this study population. In contrast, no associations were observed between metal indicators and cognitive performance or any of the other neurotrophic factors. However, we note that considering the estimated power of our findings is of 0.61, the findings must be interpreted with caution. 

In our study population, plasma copper had a modest, but significant, association with the HADS Depression scale, indicating that higher copper concentrations were associated with lower psychological distress. Copper is contained in the enzymes tyrosine hydroxylase and dopamine hydroxylase, which participate in the synthesis of neurotransmitters that play an essential role in mood [38]. To our knowledge, this is the first study to investigate the association between copper status and psychological distress in community-dwelling older adults, and therefore comparisons with other studies conducted with a similar population are precluded. When investigating individuals with depression, current literature indicates that individuals with frank depression presented with higher copper concentration in blood than controls without depression [39]. However, a sub-group analysis of the 16 studies included in the meta-analysis revealed that age was a determinant in the relationship between copper and depression, as the association was not observed for individuals aged ≥50 years [39]. Furthermore, copper concentrations in blood reported in those studies (range: 75–134 µg/dL) [39] were above what we observed in our study, and considering they could have been measured using different techniques, further comparisons are difficult. Copper concentration in foodstuffs varies according to the soil and agricultural techniques (soil enrichment, use of fungicides and bactericides, etc.) [40]. In addition to dietary intake, exposure to copper also depends on its concentration in drinking water, which is associated with groundwater and household plumbing systems. For instance, the use of copper pipelines in areas where water has acidic characteristics may result in corrosion in pipes and increase tap water copper content [41]. To our knowledge, there are no data available about copper exposure in populations from the area where this study was conducted (Victoria, Australia), and future studies should investigate different sources of the metal for this population. 

In our study population, a higher Cu/Zn ratio was associated with lower levels of psychological distress as assessed by the HADS Depression scale, although no significant association was observed for the HADS Anxiety scale or cognitive performance. No participants in this study presented with Cu/Zn ratio of >2, which has been associated with increased inflammatory response and decreased nutritional zinc status in older adults [42]. This marker reflects the balance between copper and zinc and has been positively associated with disability and mortality in elderly subjects aged 70 years and above [23]. Rather than a simple nutritional indicator, Cu/Zn ratio is mostly associated with oxidative stress and inflammatory response [23,43]. Since serum zinc was not associated with HADS Depression scale, it suggests that copper was driving the association. Therefore, our findings corroborate the notion that copper may play an important role in maintaining brain health by acting as a cofactor for neurotransmitter synthesis [38]. 

We also observed a moderate positive association between copper and BDNF concentrations, while zinc and Cu/Zn ratios were not associated with any neurotrophic factors. BDNF is part of a class of proteins called neurotrophins that are involved in survival, growth and plasticity of several cell types, including neurons [44]. Considering the importance of neurotrophins for brain health maintenance, research has linked lower concentrations of BDNF with neurodegenerative diseases [45] and depression [46]. However, no association between serum BDNF and cognitive performance was observed in healthy older adults [47], suggesting that changes in BDNF do not precede cognitive decline, as they may occur contemporaneously with decline [47]. Zinc and copper play an essential role in the synthesis and functioning of BDNF. These two metals up-regulate matrix metalloproteinases, which promote the maturation of pro-BDNF to BDNF [48,49], and enhance the cascade initiated by the neurotrophin nerve growth factor on the increment of BDNF expression [50,51]. Additionally, copper favorably regulates the activity of neurotrophins by inhibiting tyrosine phosphatase activity [51]. To our knowledge, this is the first study to investigate the association between circulating concentrations of copper and BDNF in community-dwelling older adults. These findings align with the association between plasma copper concentrations and depressive symptoms, and the tendency for better cognitive performance, as seen for BRI (despite the lack of significance) in our study population. Considering that this was a secondary analysis and therefore no power calculation was performed prior to our study, we recommend that further research aims to elucidate whether this association remains positive in cognitively intact populations and in individuals with higher concentrations of plasma copper than those found in our study.

We found no association between copper status and cognitive performance, although as noted a non-significant tendency has been seen for BRI. These findings validate the concept of copper homeostasis, which considers the essentiality of copper as a cofactor for metalloenzymes that participate in neurotransmitter synthesis [39,52]. Similarly, a previous cross-sectional study with cognitively-intact community-dwelling older adults demonstrated that women with a high concentration of copper in plasma (>215 µg/dL) had poorer cognitive function than women with low plasma copper concentration (<90 µg/dL) [53]. However, no cognitive differences between low and medium (≥90–215 µg/dL) plasma copper concentrations were observed. In men, an inverse U-shaped association between cognitive performance and copper status was observed, where both very low and very high concentrations of copper were associated with poorer cognitive performance on tests of long-term memory, calculation, and visuomotor attention, compared to intermediate circulating copper concentrations [53]. In alignment with our findings, this study corroborates the notion that copper is essential for the synthesis of neurotransmitters important for the maintenance of cognitive function. Although our study aimed to investigate cognitively-intact older adults, we highlight the importance of noting that the direction of the association between copper status and cognitive performance in individuals with Alzheimer’s disease, the main form of dementia, seems to differ from subjects without the disease. A meta-analysis reported increased copper concentrations in serum/plasma of Alzheimer’s disease patients when compared with healthy controls [20]. However, it must be considered that copper concentrations in the participants included in the meta-analysis were higher (lowest mean in the control group: 70 µg/dL) [20] than those we found in our population, and therefore it remains unclear whether such a difference between Alzheimer’s and controls in regards to copper status would exist in populations with lower concentrations. Similarly to what was observed for copper and zinc, Cu/Zn ratio was not associated with cognitive outcomes, although a non-significant tendency has been observed for GEC. Gonzalez–Dominguez et al. [24] have previously reported that individuals with Alzheimer’s disease presented with a higher Cu/Zn ratio than healthy controls. However, the results of another study conducted with Italian older adults suggested that the association between cognition and Cu/Zn ratio may not be accurate when the ratio is below the cut-off of >2 associated with increased inflammation [23], as seen in our study population.

When investigating the association between copper status and cognition, studies indicate that rather than toxicological exposure to copper, a failure of endogenous regulatory mechanisms can lead to decreased transport of copper to the brain [17]. This is linked to an increased concentration of non-bound ceruloplasmin (non-Cp) copper in plasma along with a decrease in copper bound to ceruloplasmin (Cp), which is less prone to oxidative reaction [17]. In support of this notion, several studies have reported that non-Cp copper is higher in individuals with mild cognitive impairment (MCI) and Alzheimer’s disease when compared with healthy controls [54,55,56]. Furthermore, Squitti et al. [57] demonstrated that non-Cp copper was the only significant predictor of clinical conversion from MCI toward Alzheimer’s disease. In this current study, only total copper, which encompasses both Cp and non-Cp, was measured, therefore precluding further exploration between copper status and cognition. 

In our study, zinc deficiency was present in 15.8% of men and 22.3% of women, which aligns with previous reports in healthy community-dwelling older adults from Germany [58] and Australia [59]. Older adults are at an increased risk for zinc deficiency due to age-related changes in various physiological functions such as decreased intestinal absorption and alteration in zinc transporter proteins, drug interactions, and low dietary zinc intake [37,60], which can increase the risk of senile dementia [19,20] and depression [61,62]. No associations between serum zinc and cognitive outcomes or neurotrophic factors were observed in this population. This may be explained, at least in part, by our homogenous sample of participants included due to the rigorous selection criteria, which may have resulted in a cohort ‘too cognitively healthy’ thus precluding any significant effect that zinc may have on cognitive function. Although we have not investigated the association between exposure to zinc and cognitive outcomes in this study, our findings align with previous studies that do not present conclusive evidence to recommend a modification of zinc intake to reduce the risk of Alzheimer’s disease, as neither zinc supplementation nor dietary zinc intake appear to be associated with cognitive changes in older adults with or without dementia [19]. Furthermore, we highlight that serum zinc, despite being one of the most used biomarkers to assess zinc status, presents limited sensitivity and generally does not respond to dietary zinc consumption in individuals without severe deficiency, which indicates a refined homeostatic control that uses the pools of zinc to counteract fluctuation in the serum [63,64].

A strength of this study is the inclusion of the Cu/Zn ratio along with plasma copper and serum zinc biomarkers, and the assessment of neurotrophic factors. A limitation of this study is that no information on the intake of zinc and copper via diet or supplements was considered in the data analysis. Additionally, only total copper was measured in plasma, and concentrations of Cp copper and non-Cp were not investigated. We acknowledge that the restricted eligibility criteria may have resulted in a relatively healthy population of older adults, and therefore the generalizability of these findings to all older adults is precluded. Furthermore, as this is a secondary analysis of data from a previous study limited to a cross-sectional analysis, power calculations were not performed prior to the study, and causality may not be inferred. 

## 5. Conclusions

In summary, in a sample of community-dwelling older Australian adults, greater plasma copper and higher Cu/Zn ratios were associated with lower psychological distress. Additionally, plasma copper was positively associated with BDNF concentration, a neurotrophic factor that is known to play an important role in brain plasticity. In contrast, serum zinc was not associated with cognitive performance, psychological distress, or neurotrophic factors. 

## Figures and Tables

**Figure 1 nutrients-11-02503-f001:**
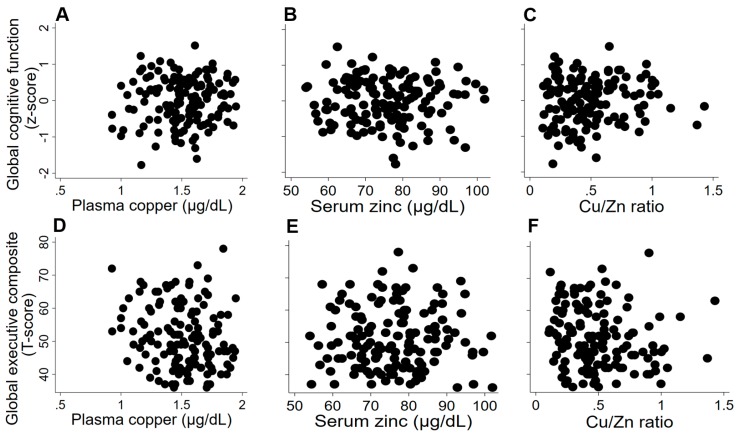
Association between Cogstate global cognitive function (**A**)–(**C**) and the BRIEF-A global executive composite (**D**)–(**F**) with copper (log-transformed value) (**A**,**D**), zinc (adjusted for IL-6) (**B**,**E**) and Cu/Zn ratio (**C**,**F**). Results were adjusted for age, sex, BMI, habitual physical activity (kJ weekly spent in vigorous activity), genotypes for APOE (carriers of ε4/non-carriers), education (primary school/high school/technical certificate/university), and history of CVD (yes/no).

**Figure 2 nutrients-11-02503-f002:**
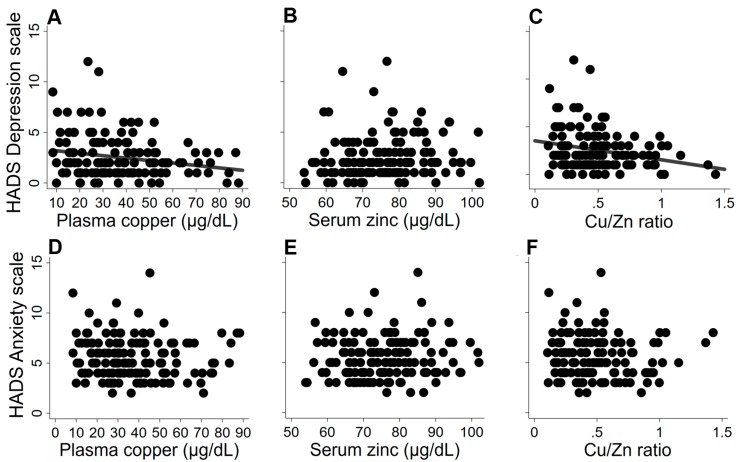
Association between HADS Depression scale (**A**)–(**C**) and HADS Anxiety scale (**D**)–(**F**) with (**A**,**D**), zinc (adjusted for IL-6) (**B**,**E**) and Cu/Zn ratio (**C**,**F**). Results were adjusted for age, sex, BMI, habitual physical activity (kJ weekly spent in vigorous activity), genotypes for APOE (carriers of ε4/non-carriers), education (primary school/high school/technical certificate/university), and history of CVD (yes/no). HADS, Hospital Anxiety and Depression Scale.

**Table 1 nutrients-11-02503-t001:** Characteristics of study population (*n* = 139).

Characteristics	Values
Age (y)	70.8 ± 4.3
Women, *n* (%)	87 (62.6)
BMI, kg/m^2^	27.8 ± 5.4
Education	
Primary/high school, *n* (%)	54 (38.9)
Technical certificate, *n* (%)	22 (15.8)
University, *n* (%)	63 (45.3)
APOE 4 carrier, *n* (%)	31 (22.3)
History of CVD, *n* (%)	17 (12.2)
Habitual physical activity, kJ/week	6070 (2326, 11765)
Metals	
Plasma copper, µg/dL	32.8 (23.0, 49.2)
Serum zinc, µg/dL ^a^	75.8 ± 10.6
Copper/Zinc ratio ^b^	0.45 (0.29, 0.64)
Neurotrophic factors	
Serum IGF-1, nmol/L	16.0 (13.5, 20.2)
Serum BDNF, ng/mL	20.0 (15.3, 25.5)
Serum VEGF, pg/mL	297.1 (170.9, 561.6)

^a^ Value adjusted for IL-6; ^b^ calculated as plasma copper divided by adjusted serum zinc. Values are presented as mean ± standard deviation, *n* (%), or median (25th, 75th percentile). APOE, apolipoprotein E; BDNF, brain-derived neurotrophic factor; BMI, body mass index; CVD, cardiovascular disease; IGF-1, insulin-like growth factor 1; VEGF, vascular endothelial growth factor.

**Table 2 nutrients-11-02503-t002:** Associations between plasma copper, serum zinc, and Cu/Zn ratio and cognitive outcomes and HADS scales in older adults (*n* = 139).

	Plasma Copper			Serum Zinc			Cu/Zn ratio		
Outcomes	β	95% CI	*p*	β	95% CI	*p*	β	95% CI	*p*
Cogstate									
Global cognitive function ^a^	0.001	−0.004, 0.007	0.595	−0.005	−0.01, 0.00	0.314	0.15	−0.24, 0.55	0.453
Working memory/learning ^a^	0.000	−0.007, 0.007	0.999	−0.003	−0.01, 0.01	0.612	0.05	−0.45, 0.56	0.835
Attention/psychomotor ^a^	0.002	−0.006, 0.010	0.579	−0.003	−0.02, 0.01	0.638	0.14	−0.43, 0.71	0.624
BRIEF-A									
Global executive composite ^b^	−0.079	−0.166, 0.007	0.073	0.05	−0.11, 0.20	0.550	−6.00	−12.16, 0.16	0.056
Behavioral regulation index ^b^	−0.085	−0.171, 0.001	0.053	0.01	−0.15, 0.16	0.926	−5.71	−11.88, 0.45	0.069
Metacognition index ^b^	−0.063	−0.158, 0.031	0.186	0.07	−0.10, 0.23	0.422	−5.37	−12.10, 1.36	0.117
HADS									
Depression scale	−0.024	−0.044, −0.004	0.019	0.02	−0.01, 0.06	0.220	−1.99	−3.41, −0.57	0.006
Anxiety scale	−0.005	−0.025, 0.015	0.609	0.01	−0.02, 0.04	0.572	−0.54	−1.97, 0.89	0.456

^a^*z*-score; ^b^ t-score; β represent unstandardized beta-coefficients. Results were adjusted for age, sex, BMI, habitual physical activity (kJ weekly spent in vigorous activity), genotypes for APOE (carriers of ε4/non-carriers), education (primary/high school/technical certificate/university), and history of CVD (yes/no). BRIEF-A, Behavior Rating of Inventory Executive Function-Adult version; HADS, Hospital Anxiety and Depression Scale.

**Table 3 nutrients-11-02503-t003:** Associations between plasma copper, serum zinc and Cu/Zn ratio with neurotrophic factors in older adults (*n* = 139).

	Plasma Copper			Serum Zinc			Cu/Zn Ratio		
Neurotrophic factors	β	95% CI	*p*	β	95% CI	*p*	β	95% CI	*p*
BDNF (ng/mL) ^a^	0.004	0.001; 0.007	0.021	0.005	0.00; 0.01	0.062	0.21	−0.02; 0.44	0.080
IGF-1 (nmol/L) ^a^	0.000	–0.003; 0.002	0.875	0.004	0.00; 0.01	0.065	–0.05	–0.24; 0.13	0.550
VEGF (pg/mL) ^a^	0.003	–0.005; 0.012	0.411	0.010	–0.04; 0.02	0.174	0.15	–0.44; 0.76	0.634

^a^ log-transformed; β represent unstandardized beta-coefficients. Results were adjusted for age, sex, BMI, habitual physical activity (kJ weekly spent in vigorous activity), genotypes for APOE (carriers of ε4/non-carriers), and history of CVD (yes/no). BDNF, brain-derived neurotrophic factor; IGF-1, insulin-like growth factor 1; VEGF, vascular endothelial growth factor.

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
