# Peer review of "Greater Circulating Copper Concentrations and Copper/Zinc Ratios are Associated with Lower Psychological Distress, But Not Cognitive Performance, in a Sample of Australian Older Adults"

_nutrients, 2019, doi:10.3390/nu11102503_

Round 1
Reviewer 1 Report
This is a soundly performed and well written-up analysis on a relatively interesting topic. Perhaps most importantly, the authors are aware of the limitations of their study's results significance. They clearly state these limitations, without over-interpretation of the results. As cognitive decline and emotional dysregulation are common problems in ever-expanding aging population, without clearly efficient therapeutic measures, any information on the potential effect of modifiable life variables, such as specific nutrients intake, should be of interest.
Author Response
Thank you for your comments.
Reviewer 2 Report
Mravunac and colleagues take advantage of a baseline database of the Seniors’ Thinking, Exercise and Protein Study (STEPS) which uses a community-dwelling older Australian healthy adults. I believe because the subjects are all healthy in terms of depression and cognition deficits, the entire discussion/conclusion should be assessed. The title also does not reflect the study as depressive symptoms and cognitive impairment were found in the subject tested.
The study still has its value indeed, in providing a correlation of Zn/Cu and different scores in cognitive/depression test but not associated with the diseases per se.
Other comments:
Because of the nature of the study, it is not possible to know if the participants take alimentary supplementation containing Zn/Cu and thus, not possible to find differences in supplemented and naturally observed Zn/Cu in food. However, it would nice to have a brief discussion of the role of supplemented Zn/Cu in depression and cognition compared to Zn/Cu observed in food (if any). Also, the role of the Zn/Cu originated from pollution could be mentioned.
Show main data in figure format instead of a table would be more visually appealing.
L288, please more clear with the "BDNF and cognitive performance in healthy subjects is conflicting". What the studies are shown?
Maybe split men and women data?
L312/313: brain health cannot be defined by depressive and cognitive scores only, especially if the score does not show depression/clear cognitive impairment.
L355: a power test should be performed with the data available to demonstrate how strong/weak is the data.
Author Response
Comment 1: I believe because the subjects are all healthy in terms of depression and cognition deficits, the entire discussion/conclusion should be assessed.
Because evidence indicates that older adults are at a higher risk of dementia and depression, we find it relevant to compare and contrast the zinc and copper status in individuals with and without these diseases. We have clearly characterised our study population at the beginning of the discussion and in the conclusion, and clarified in the text when comparing studies conducted in different populations as follows:
“The main finding from this study was that in a sample of community-dwelling older adults free of dementia and depression, greater plasma copper, as well as Cu/Zn ratio, were associated with lower depressive symptoms.” (lines 274-276)
“In summary, in a sample of healthy community-dwelling older Australian adults free of dementia and depression, greater plasma copper and higher Cu/Zn ratio were associated with lower depressive symptoms.” (lines 405-407)
“To our knowledge, this is the first study to investigate the association between copper status and symptoms of depression and anxiety in older adults free of depression, and therefore comparisons with other studies are precluded. Current literature on copper and depression indicates that individuals with depression presented with a higher copper concentration in blood than controls without depression [1].” (lines 284-288)
“Although our study aimed to investigate older adults free of dementia, we highlight the importance of noting that the direction of the association between copper status and cognitive performance in individuals with Alzheimer’s disease, the main form of dementia, seems to differ from subjects without the disease.” (lines 349-353)
Comment 2: The title also does not reflect the study as depressive symptoms and cognitive impairment were found in the subject tested.
We have modified the title to “Greater circulating copper concentrations and copper/zinc ratio are associated with lower depressive symptoms, but not cognitive performance in a sample of Australian older adults” to better reflect the main findings of the manuscript.
Comment 3: Because of the nature of the study, it is not possible to know if the participants take alimentary supplementation containing Zn/Cu and thus, not possible to find differences in supplemented and naturally observed Zn/Cu in food. However, it would nice to have a brief discussion of the role of supplemented Zn/Cu in depression and cognition compared to Zn/Cu observed in food (if any). Also, the role of the Zn/Cu originated from pollution could be mentioned.
We wish to clarify that the aim of this study was to investigate the association between status of copper and zinc, and not necessarily the intake of these nutrients. Therefore, comparing the consumption of supplements with naturally occurring food sources is not pertinent for the purpose of this study. However, as suggested by the reviewer, we have included information about sources of copper (diet and environment) and impact of zinc consumption (via diet or supplements) on the risk of dementia.
“Copper concentration in foodstuffs varies according to the soil and agricultural techniques (soil enrichment, use of fungicides and bactericides, etc) [2]. In addition to dietary intake, exposure to copper also depends on its concentration in drinking water, which is associated with groundwater and household plumbing systems. For instance, the use of copper pipelines in areas where water has acidic characteristics may result in corrosion in pipes and increase tap water copper content [3]. To our knowledge, there are no data available about copper exposure in populations from the area where this study was conducted (Victoria, Australia), and future studies should investigate different sources of the metal for this population.” (lines 294-301)
“Although we have not investigated the association between exposure to zinc and cognitive outcomes in this study, our findings align with previous studies that do not present conclusive evidence to recommend a modification of zinc intake to reduce the risk of Alzheimer’s disease, as neither zinc supplementation nor dietary zinc intake appear to be associated with cognitive changes in older adults with or without dementia [4]. Furthermore, we highlight that serum zinc, despite being one of the most used biomarkers to assess zinc status, presents limited sensitivity and generally does not respond to dietary zinc consumption in individuals without severe deficiency, which indicates a refined homeostatic control that uses the pools of zinc to counteract fluctuation in the serum [5, 6].” (lines 386-394)
Comment 4: Show main data in figure format instead of a table would be more visually appealing.
As per the reviewer’s suggestion, we are now displaying the findings for global cognitive function, global executive function and HADS composites in figures (Figures 1 and 2).
Comment 5: L288, please more clear with the "BDNF and cognitive performance in healthy subjects is conflicting". What the studies are shown?
We have clarified the details of the study as follows:
“However, no association between serum BDNF and cognitive performance was observed in healthy older adults [7], suggesting that changes in BDNF do not precede cognitive decline, as they may occur contemporaneously with decline [7]” (lines 320-323)
Comment 6: Maybe split men and women data?
We tested whether there was a sex interaction for the associations between cognitive outcomes and neurotrophic factors (dependent variables) with the metal markers in multivariable models adjusted for covariates as described in the manuscript (cognitive outcomes: age, BMI, habitual physical activity, genotypes for APOE, education (primary/high school / technical certificate / university), and history of CVD; neurotrophic factors: age, BMI, habitual physical activity, genotypes for APOE, and history of CVD), and no significant interaction was observed. Therefore, results were not split by sex. We highlight, however, that all linear regression models were adjusted for sex to account for potential effect. We have added the following relevant information in the Statistical Analysis section of the manuscript as follows:
“We tested whether there were any sex interactions for the associations between cognitive outcomes and neurotrophic factors (dependent variables) with the metal markers in multivariable models adjusted for the same covariates as described above, and no significant interactions were observed.” (lines 220-223)
Comment 7: L312/313: brain health cannot be defined by depressive and cognitive scores only, especially if the score does not show depression/clear cognitive impairment.
We agree that with the reviewer that the sentence was misleading, and thus we have provided clearer information:
“In alignment with our findings, this study corroborates the notion that copper is essential for the synthesis of neurotransmitters important for maintenance of cognitive function.” (lines 347-349)
Comment 8: L355: a power test should be performed with the data available to demonstrate how strong/weak is the data.
We have calculated the F test considering the highest significant R2 observed in our findings (0.08) and the estimated power is of 0.69. We have added this information in the Statistical analysis section, and commented this result in the discussion.
“Considering this is a secondary analysis, power calculations were not performed prior to the study. However, an F test was calculated for the coefficient of determination (R2) considering the lowest significant R2 observed in our findings (0.07) and the estimated power is of 0.61.” (lines 226-229)
“Considering the estimated power of our findings is of 0.61, the findings must be interpreted with caution.” (lines 402-403)
Ni, M., et al., Copper in depressive disorder: A systematic review and meta-analysis of observational studies. Psychiatry Research, 2018. 267: p. 506-515. Thompson, T., et al., Copper levels in buccal cells of vineyard workers engaged in various activities. Ann Occup Hyg, 2012. 56(3): p. 305-14. Bost, M., et al., Dietary copper and human health: Current evidence and unresolved issues. Journal of Trace Elements in Medicine and Biology, 2016. 35: p. 107-115. Loef, M., N. von Stillfried, and H. Walach, Zinc diet and Alzheimer's disease: a systematic review. Nutr Neurosci, 2012. 15(5): p. 2-12. Johnson, P.E., et al., Homeostatic control of zinc metabolism in men: zinc excretion and balance in men fed diets low in zinc. Am J Clin Nutr, 1993. 57(4): p. 557-65. Sian, L., et al., Zinc absorption and intestinal losses of endogenous zinc in young Chinese women with marginal zinc intakes. Am J Clin Nutr, 1996. 63(3): p. 348-53. Nettiksimmons, J., et al., The associations between serum brain-derived neurotrophic factor, potential confounders, and cognitive decline: a longitudinal study. PLoS One, 2014. 9(3): p. e91339.
Round 2
Reviewer 2 Report
The manuscript has improved in quality and had made more clear. I, however still believe that the main issue of the manuscript is the way it is approached. Until I am convinced so, I believe that an adult free of dementia and depression cannot have depressive symptoms. At maximum, what the authors are trying to describe is the blues (sadness, loneliness, or grief, etc) for depressive symptoms and cognitive performance for dementia. Although say that someone is depressive is popular jargon, this is a scientific study and correct wording must be placed. This study is not talking about a disease (depression, dementia, cancer, etc) so I do see room for depressive symptom being discussed elsewhere.
In addition, because the healthy older adults do have a wider range for diseases (as stated in the selection criteria), I still think that the use of the term “free of dementia and depression” is reductionist. Some may think that the insulin intake/diabetes are as relevant that “free of depression/dementia. To note, diabetes is sometimes associated with depression and dementia.
Discussion: due the fact that the entire study is underpowered, I suggest to indicate this in the first paragraph of the discussion.
Author Response
"Please see the attachment
